# CAN ONE HEAR THE SHAPE OF A NEURAL NETWORK?: SNOOPING THE GPU VIA MAGNETIC SIDE CHANNEL

## ABSTRACT

We examine the magnetic flux emanating from a graphics processing unit's (GPU) power cable, as acquired by a cheap $3 induction sensor, and find that this signal betrays the detailed topology and hyperparameters of a black-box neural network model. The attack acquires the magnetic signal for one query with unknown input values, but known input dimension and batch size. The network reconstruction is possible due to the modular layer sequence in which deep neural networks are evaluated. We find that each layer component's evaluation produces an identifiable magnetic signal signature, from which layer topology, width, function type, and sequence order can be inferred using a suitably trained classifier and an optimization based on integer programming. We study the extent to which network specifications can be recovered, and consider metrics for comparing network similarity. We demonstrate the potential accuracy of this side channel attack in recovering the details for a broad range of network architectures, including random designs. We consider applications that may exploit this novel side channel exposure, such as adversarial transfer attacks. In response, we discuss countermeasures to protect against our method and other similar snooping techniques.

## 1 INTRODUCTION

The Graphics Processing Unit (GPU) is a favored vehicle for executing a neural network. As it computes, it also hums—electromagnetically. What can this hum tell us? Could listening to the GPU's electromagnetic (EM) radiation reveal details about the neural network? We study this question and find that magnetic induction sensing reveals a detailed network structure, including both topology and hyperparameter values, from inferences of otherwise unknown networks running on GPUs.

Reverse engineering a network structure has attracted increasing research effort, motivated by several concerns. First, it has been well known that the performance of a network model hinges on its judiciously designed structure—but finding an effective design is no easy task. Significant time and energy is expended in searching and fine-tuning network structures (Zoph et al., 2018). Moreover, in industry, optimized network structures are often considered confidential intellectual property. It is therefore important to understand the extent to which this valuable, privileged information can be compromised.

Worse yet, a reverse engineered "surrogate" model also makes the black-box "victim" model more susceptible to adversarial *transfer attacks* (Papernot et al., 2017; Liu et al., 2016), in which a vulnerability identified in the surrogate is exploited on the victim. Success in the exploit is contingent on the ability of the surrogate to successfully model the vulnerabilities of the victim. Recovering accurate, detailed network topology and hyperparameters informs the modeling of a good surrogate.

We examine the fluctuation of magnetic flux from the GPU's power cable, and ask whether a passive observer can glean the information needed to reconstruct neural network structure. Remarkably, we show that, through magnetic induction sensing, a passive observer can reconstruct the *complete* network structure even for *large* and *deep* networks.

**Threat model.** We consider an adversary that **(i)** is able to place a magnetic induction sensor in close proximity to the GPU's power cable, **(ii)** knows the dimension of the input feature vector, and **(iii)** is able to launch a query of known batch size. We also consider that our attacker uses the same deep learning framework (e.g., PyTorch, TensorFlow) as the black-box model. The adversary is otherwise weak, lacking access to the model source, binaries, training data, and underlying training data distribution; without ability to execute code on the host CPU and GPU; and without knowledge of the input values and output results of the launched queries. Not only that—it also lacks direct

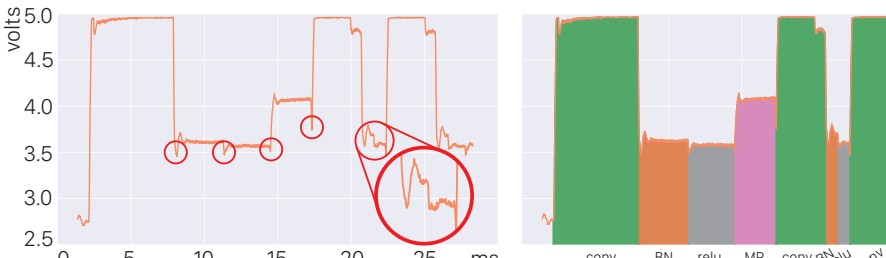

Figure 1: **Leaked magnetic signal.** (left) Our induction sensor captures a magnetic signal when a CNN is running on the GPU. We observe strong correlation between the signal and the network steps. Across two steps, the GPU has to synchronize, resulting in a sharp drop of the signal level (highlighted by selected red circles). (right) We can accurately classify the network steps and reconstruct the topology, as indicated by the labels under the $x$-axis. Here we highlight the signal regions associated with convolutions (conv), batch-norm (BN), Relu non-linear activations (relu), max-pooling (MP), and adding steps together (add).

access to the GPU hardware, beyond the proximity to the power cable. The adversary only requires access to their own GPU hardware, matching the brand/version of the victim, e.g., as purchased on the open market.

**Physical principle.** The GPU consumes energy at a variable rate that depends on operations performed. Every microprocessor instruction is driven by transistor electron flows, and different instructions require different power levels (Grochowski & Annavaram, 2006). The many compute cores of a GPU amplify the fluctuation in energy consumption, and so too the current drawn from the power cable. Current induces magnetic flux governed by the Biot-Savart law (Griffiths, 2005), and current fluctuations induce EM ripples whose propagation through the environment is governed by the Ampère-Maxwell law. Even a cheap, $3 magnetic induction sensor (see Fig. 2) placed within a few millimeters of the power cable suffices to record these EM ripples.

**Technique and results.** To reconstruct the black-box network's structure, we propose a two-step approach. First, we estimate the network topology, such as the number and types of layers, and types of activation functions, using a suitably trained neural network classifier. Then, for each layer, we estimate its hyperparameters using another set of deep neural network (DNN) models. The individually estimated hyperparameters are then jointly optimized by solving an integer programming problem to enforce consistency between the layers. We demonstrate the potential accuracy of this side-channel attack in recovering the details for a wide range of networks, including large, deep networks such as ResNet101. We further apply this recovery approach to demonstrate black-box adversarial transfer attacks.

## 1.1 RELATED WORK: MODEL EXTRACTION BY QUERIES AND SIDE-CHANNEL ANALYSIS

Our work falls under the umbrella of black-box model extraction. Absent access to the model's internals, one might infer structure from observed input-output pairs. For instance, Tramèr et al. (2016) demonstrated that, for simple models such as decision trees and support vector machines hosted on a cloud, certain internal information can be extracted via a multitude of queries. This approach, which was extended to infer details of deep neural networks (Oh et al., 2019; Liu et al., 2016; Duddu & Rao, 2019), is typically able to recover certain information, such as optimization learning rate and network structure type, but has not demonstrated recovery of full structural details.

A contrasting approach, side-channel analysis (SCA), extracts information gained from the *physical* implementation of a model, rather than in the mathematical model itself. Analysis of timing (Kocher, 1996), power (Kocher et al., 1999; Luo et al., 2015), cache flushes (Yarom & Falkner, 2014), and audio (Genkin et al., 2014) have been prominently demonstrated to extract secret keys from cryptographic procedures such as the Digital Signature and Advanced Encryption Standards.

SCA was recently used to infer machine learning models by observing power consumption profiles (Xiang et al., 2020; Wei et al., 2018; Dubey et al., 2019), timing information (Duddu et al., 2018) and memory/cache access (Hu et al., 2020; Hong et al., 2019; Hua et al., 2018; Yan et al., 2020). These methods placed a malware process on the machine hosting the black-box model. Our threat model does not involve introducing processes on the host.

Recently, Batina et al. (2019) exploited EM radiation for network model extraction. They focused on EM radiation from embedded processors. In contrast to GPUs, embedded processors emit a relatively weak EM signal, necessitating delicate measurement devices and even mechanical opening of the chip package.

**Our advance.** All these previous works are demonstrated on shallow networks (e.g., fewer than 20 layers). It remains unclear whether these methods can also extract deep network models, ones that are structurally more complex and more prevalent in practice. We demonstrate successful recovery of the full structure of deep networks, such as RetNet101 (He et al., 2016). With that, we hope to raise awareness of the GPU's EM radiation as an information-rich, easily-probed side channel.

## 2 MAGNETIC SIGNALS FROM GPUs

Before getting into the weeds, let us provide our intuition: we think of the magnetic signal as the GPU's "speech." The GPU speaks a series of "words," demarcated by silence. Each word names the computational *step* that was executed. Let us now refine this explanation further, and ground it in physical principles.

We use *step* to refer to performing a specific kind of network operation, such as a linear operation, batch normalization, pooling, activation function, etc. A *layer* is a sequences of steps, e.g., a **(i)** linear operation, then **(ii)** pooling, then **(iii)** activation. While there may be data dependencies between steps, there are no such dependencies within a step.

The parallel nature of GPU computation lends itself to a natural implementation of networks, wherein each step is executed in parallel, i.e., single instruction multiple data (SIMD) parallelism. Transitions between steps, however, are synchronized (Buck, 2007): in our example above, activation begins only after pooling completes. This cross-step synchronization allows for implementations structured into modules, or GPU *kernels*. This modular approach is employed in widely-used deep learning frameworks such as PyTorch and TensorFlow (Paszke et al., 2019; Abadi et al., 2016).

**Signal.** Kernel execution demands transistor flips, which place electric load on the GPU processor, in turn emitting magnetic flux from its power cable. An induction sensor measures this flux and produces proportional voltage. The time-varying voltage is our acquired *signal* (see Fig. 1).

Different steps correspond to different GPU kernels, transistor flips, electric loads, and signal characteristics, which are distinguished even by the naked eye (see Fig. 1). Cross-step synchronization involves idling, dramatically reducing electric load and signal level (see Fig. 1). These rapid sharp drops demarcate steps.

We observe that signal strongly correlates to the *kind* of GPU operations, rather than the specific *values* of computed floating point numbers. We verify this by examining signals using both PyTorch and TensorFlow and on multiple kinds of GPUs (see Sec. 5).

The signal is also affected by the input to the network. Although the specific input data values do not influence the signal, the input data size does. When the GPU launches a network, the size of its single input (e.g., image resolution) is fixed. But the network may be provided with a batch of input data (e.g., multiple images). As the batch size increases, more GPU cores will be utilized in each step. The GPU consequently draws more power, which in turn strengthens the signal. Once all GPU cores are involved, further increase of input batch size will not increase the signal strength, but elongate the execution time until the GPU runs out of memory.

Therefore, in launching a query to the black-box network model, the adversary should choose a batch size sufficiently large to activate a sufficient number of GPU cores to produce a sufficient signal-to-noise ratio. We find that the range of the proper batch sizes is often relatively large (e.g., $64 \sim 96$ for ImageNet networks), loosely depending on the size of the single input's features and the parallel computing ability of the GPU. In practice, the adversary can choose the batch size by experimenting with their own GPUs under various image resolutions.

Notably however, we do not require knowledge of batch size to robustly recover network topology (as opposed to hyperparameters), only that the batch size is sufficiently large enough to provide a clear signal. While we used a consumer friendly sensor with limited sampling rate (see 4.1) and corresponding signal-to-noise ratio (SNR), a sensor with high sampling rate and SNR would correspondingly require a smaller minimum batch size.

# 3 Signal Analysis and Network Reconstruction

We prepare for the attack by training several *recovery* DNN models; we refer to training before the attack as *pre*training. After the attacker launches a batch query (whose input and output values are irrelevant) we recover structure from the acquired signal in two stages: **(i)** topology and **(ii)** hyperparameters. To recover topology, a pretrained DNN model associates a *step* to every signal *sample*. This per-sample classification partitions the signal into segments corresponding to steps. We estimate hyperparameters for each *individual* segment in isolation, using a step-specific pretrained DNN model, and resolve inconsistencies between consecutive segments using an integer program. The pretraining of our recovery DNN models is hardware-specific, and good recovery requires pretraining on like hardware.

## 3.1 Topology Recovery

Bidirectional Long Short Term Memory (biLSTM) networks are well-suited for processing time-series signals (Graves et al., 2005). We train a biLSTM network to classify each signal sample $s_i$ predicting the step $C(s_i)$ that generated $s_i$ (see Fig. 1-b). The training dataset consists of annotated signals constructed automatically (see Sec. 4.2). We train the biLSTM by minimizing the standard cross-entropy loss between the predicted per-sample labels and the ground-truth labels (see Appx. A for details). By identifying the sequence of steps, we recovered the layers of the network, including their *type* (e.g., fully connected, convolution, recurrent, etc.), activation function, and any subsequent forms of pooling or batch normalization. What remains is to recover layer hyperparameters.

## 3.2 Hyperparameter Estimation

**Hyperparameter consistency.** The number of hyperparameters that describe a layer type depends on its linear step. For instance, a CNN layer type's linear step is described by size, padding, kernel size, number of channels, and stride hyperparameters. Hyperparameters within a layer must be *intra-consistent*. Of the six CNN hyperparameters (stride, padding, dilation, input, output, and kernel size), any one is determined by the other five. Hyperparameters must also be *inter-consistent* across consecutive layers: the output of one layer must fit the input of the next. A brute-force search of consistent hyperparameters easily becomes intractable for deeper networks; we therefore first estimate hyperparameters for each layer in isolation, and then jointly optimize to obtain consistency.

**Initial estimation.** We estimate a specific hyperparameter of a specific layer type, by pretraining a DNN. We pretrain a suite of such DNNs, one for each (layer type, hyperparameter) pairing. Once the layers (and their types) are recovered, we estimate each hyperparameter using these pretrained (layer type, hyperparameter) recovery DNNs.

Each DNN is comprised of two 1024-node fully connected layers with dropout. The DNN accepts two (concatenated) feature vectors describing two signal segments: the linear step and immediately subsequent step. That subsequent step (e.g., activation, pooling, batch normalization) tends to require effort proportional to the linear step's output dimensions, thus its inclusion informs the estimated output dimension. Each segment's feature vector is assembled by **(i)** partitioning the segment uniformly into $N$ windows, and computing the average value of each window, **(ii)** concatenating the time duration of the segment. The concatenated feature vector has a length of $2N + 2$.

The DNN is trained with our automatically generated dataset (see Sec. 4.2). The choice of loss function depends on the hyperparameter type: For a hyperparameter drawn from a wide range, such as a *size*, we minimize mean squared error between the predicted size and the ground truth (i.e., regression). For a hyperparameter drawn from a small discrete distribution, such as *stride*, we minimize the cross-entropy loss between the predicted value and the ground truth (i.e., classification). In particular, we used regression for sizes, and classification for all other parameters.

**Joint optimization.** The initial estimates of the hyperparameters are generally not *fully* accurate, nor consistent. To enforce consistency, we jointly optimize all hyperparameters, seeking values that *best fit their initial estimates, subject to consistency constraints*. Our optimization minimizes the convex quadratic form

$$\min_{x_i \in \mathbb{Z}^{0+}} \sum_{i \in \mathcal{X}} (x_i - x_i^*)^2 \ , \quad \text{subject to consistency constraints,} \tag{1}$$

where $\mathcal{X}$ is the set of all hyperparameters across all layers; $x_i^*$ and $x_i$ are the initial estimate and optimal value of the $i$-th hyperparameter, respectively. The imposed consistency constraints are:

Figure 2: Placement of the magnetic induction sensor on the power cord works regardless of the GPU model, providing a common weak-spot to enable current-based magnetic side-channel attacks.

(i) The output size of a layer agrees with the input size of the next next layer.

(ii) The input size of the first layer agrees with the input feature size.

(iii) The output size of a CNN layer does not exceed its input size (due to convolution).

(iv) The hyperparameters of a CNN layer satisfy

$$s_{\text{out}} = \left\lfloor \frac{s_{\text{in}} + 2\beta - \gamma(k-1) - 1}{\alpha} + 1 \right\rfloor, \tag{2}$$

where $\alpha$, $\beta$, $\gamma$, and $k$ denote the layer's stride, padding, dilation, and kernel size, respectively.

(v) Heuristic constraint: the kernel size must be odd.

Among these constraints, **(i-iii)** are linear constraints, which preserves the convexity of the problem. The heuristic **(v)** can be expressed as a linear constraint: for every kernel size parameter $k_j$, we introduce a dummy variable $\tau_j$, and require $k_j = 2\tau_j + 1$ and $\tau_j \in \mathbb{Z}^{0+}$. Constraint **(iv)**, however, is troublesome, because the appearance of kernel size $\alpha$ and dilation $\gamma$, both of which are optimization variables, make the constraint nonlinear.

Since all hyperparameters are non-negative integers, the objective must be optimized via integer programming: IP in general case is NP-complete (Papadimitriou & Steiglitz, 1998), and the non-linear constraint **(iv)** does not make life easier. Fortunately, both $\alpha$ and $\gamma$ have very narrow ranges in practice: $\alpha$ is often set to be 1 or 2, and $\gamma$ is usually 1, and they rarely change across all CNN layers in a network. As a result, they can be accurately predicted by our DNN models; we therefore retain the initial estimates and do not optimize for $\alpha$ and $\gamma$, rendering (2) linear. Even if DNN models could not reliably recover $\alpha$ and $\gamma$, one could exhaustively enumerate the few possible $\alpha$ and $\gamma$ combinations, and solve the IP problem (1) for each combination, and select the best recovery.

The IP problem with a quadratic objective function and linear constraints can be easily solved, even when the number of hyperparameters is large (e.g., $> 1000$). In practice, we use IBM CPLEX (Cplex, 2009), a widely used IP solver. Optimized hyperparameters remain close to the initial DNN estimates, and are guaranteed to define a valid network structure.

## 4 EXPERIMENTAL SETUP

### 4.1 HARDWARE SENSORS

We use the DRV425 fluxgate magnetic sensor from Texas Instruments for reliable high-frequency sensing of magnetic signals (Instruments, 2020; Petrucha & Novotny, 2018). This sensor, though costing only $3 USD, outputs robust analog signals with a 47kHz sampling rate and $\pm 2$mT sensing range. For analog to digital conversion (ADC), we use the USB-204 Digital Acquisition card, a 5-Volt ADC from Measurement Computing (Computing, 2020). This allows a 12-bit conversion of the signal, mapping sensor readings from -2mT$\sim$2mT to 0V$\sim$5V.

### 4.2 DATASET CONSTRUCTION

**Sensor placement.** To avoid interference from other electric components, we place the sensor near the GPU's magnetic induction source, anywhere along the power cable. Because magnetic flux decays inversely proportional to the squared distance from the source, according to the Biot-Savart law (Griffiths, 2005), we position the sensor within millimeters of the cable casing (see Fig. 2).

**Data capture.** Pretraining the recovery DNN models (recall Sec. 3) requires an annotated dataset with pairwise corresopndence between signal and step types (see Fig. 2). We can automatically generate an annotated signal for a given network and specific GPU hardware, simply

by executing a query (with arbitrary input values) on the GPU to acquire the signal. Times-tamped ground-truth GPU operations are made available by most deep learning libraries (e.g., `torch.autograd.profiler` in PyTorch and `tf.profiler` in TensorFlow). A difficulty in this process lies in the fact that the captured (47kHz) raw signals and the ground truth GPU traces run on different clocks. Similar to the use of clapperboard to synchronize picture and sound in filmmaking, we precede the inference query with a short intensive GPU operation to induce a sharp spike in the signal, yielding a synchronization landmark (see Fig. S3). We implemented this "clapperboard" by filling a vector with random floating point numbers.

**Training Set.** The set of networks to be annotated could in principle consist solely of randomly generated networks, on the basis that data values and "functionality" are irrelevant to us, and the training serves to recover the substeps of a layer; or of curated networks or those found in the wild, on the basis that such networks are more indicative of what lies within the black-box. We chose to construct our training set as a mixture of both of these approaches. All in all we consider $500$ networks for training leading to a total of $5708$ network steps we aim to identify. We will release the complete training and test datasets, along with source code and hardware schematics for full reproducibility.

## 5 RESULTS

This section presents the major empirical evaluations of our method. We refer the reader to Appx. B for complete results, additional experiments, and more thorough discussion.

### 5.1 ACCURACY OF NETWORK RECONSTRUCTION

**Test dataset.** We construct a test dataset fully separate from the training dataset. Our test dataset consists of $64$ randomly generated networks in a way similar to Sec. 4.2. The number of layers ranges from 30 to 50 layers. To diversify our zoology of test models, we also include smaller networks that are under 10 layers, LSTM networks, as well as ResNets (18, 34, 50, and 101). Altogether, each test network has up to 514 steps. In total, the test dataset includes 5708 network steps, broken down into $1808$ activation functions, $1975$ additional batch normalization and pooling, and $1925$ convolutional, fully connected, and recurrent layers. When we construct these networks, their input image resolutions are randomly chosen from [$224 \times 224, 96 \times 96, 64 \times 64, 48 \times 48, 32 \times 32$]: the highest resolution is used in ImageNet, and lower ones are used in datasets such as CIFAR.

**Topology reconstruction.** As discussed in Sec. 3, we use a biLSTM model to predict the network step for each single sample. Table 1 reports its accuracy, measured on an Nvidia TITAN V GPU. There, we also break the accuracy down into measures of individual types of network steps, with an overall accuracy of **96.8%**. An interesting observation is that the training and test datasets are both unbalanced in terms of signal samples (see last column of Table 1). This is because in practice convolutional layers are computationally the most expensive, while activation functions and pooling are lightweight. Also, certain steps like average pooling are much less frequently used. While such data imbalance does reflect reality, when we use them to train and test, most of the misclassifications occur at those rarely used, lightweight network steps, whereas the majority of network steps are classified correctly.

Table 1: Classification accuracy of network steps on TITAN V.

| Layer Type | Prec. | Rec. | F1 | # samples |
|---|---|---|---|---|
| LSTM | .997 | .992 | .995 | 8704 |
| Conv | .993 | .996 | .994 | 447968 |
| Fully-connected | .901 | .796 | .846 | 10783 |
| Add | .984 | .994 | .989 | 22714 |
| BatchNorm | .953 | .955 | .954 | 47440 |
| MaxPool | .957 | .697 | .806 | 4045 |
| AvgPool | .371 | .760 | .499 | 675 |
| ReLU | .861 | .967 | .911 | 28512 |
| ELU | .464 | .825 | .594 | 2834 |
| LeakyReLU | .732 | .578 | .646 | 9410 |
| Sigmoid | .694 | .511 | .588 | 8744 |
| Tanh | .773 | .557 | .648 | 4832 |
| Weighted Avg. | **.968** | **.967** | **.966** | - |

We evaluate the quality of topology reconstruction using normalized Levenshtein distance (i.e., one of the edit distance metrics) that has been used to evaluate network structure similarity (Graves et al., 2006; Hu et al., 2020). Here, Levenshtein distance measures the minimum number of operations—including adding/removing network steps and altering step type—needed to fully rectify a recovered topology. This distance is then normalized by the total number of steps of the target network.

We report the detailed results in Fig. S2 in the appendix. Among the 64 tested networks, 40 of the reconstructed networks match precisely their targets, resulting in *zero* Levenshtein distance. The

Table 2: Model extraction accuracy on CIFAR-10 across different GPUs.

| Model | Target | Titan-V | Titan-X | GTX-1080 | GTX-960 |
|---|---|---|---|---|---|
| VGG-11 | 89.03 | 89.61 | 89.63 | 88.46 | 88.3 |
| VGG-16 | 90.95 | 91.08 | 91.03 | 89.33 | 90.78 |
| AlexNet | 81.68 | 85.26 | 85.11 | 85.27 | 85.03 |
| ResNet-18 | 92.77 | 92.61 | 92.82 | 92.79 | 92.04 |
| ResNet-34 | 92.21 | 92.28 | 92.95 | 90.81 | 92.71 |
| ResNet-50 | 90.89 | 91.8 | 91.97 | 91.2 | 91.29 |
| ResNet-101 | 91.58 | 91.91 | 91.85 | 91.37 | 91.72 |

average normalized Levenshtein distance of all tested networks is **0.118**. To provide a sense of how the Levenshtein distance is related to the network's ultimate performance (i.e., its classification accuracy), we conduct an additional experiment and discuss it in Appx. B.1.

**DNN hyperparameter estimation.**   Next, we report the test accuracies of our DNN models (discussed in Sec. 3.2) for estimating hyperparameters of convolutional layers. Our test data here consists of 1804 convolutional layers. On average, our DNN models have **96~97%** accuracy. The break-down accuracies for individual hyperparameters are shown in Table S3 of the appendix.

**Reconstruction quality measured as classification accuracy.**   Ultimately, the reconstruction quality must be evaluated by how well the reconstructed network performs in the task that the original network aims for. To this end, we test seven networks, including VGGs, AlexNet, and ResNets, that have been used for CIFAR-10 classification (shown in Table 2). We treat those networks as black-box models and reconstruct them from their magnetic signals. We then train those reconstructed networks and compare their test accuracies with the original networks' performance. Both the reconstructed and original networks are trained with the same training dataset for the same number of epochs. The results in Table 2 show that for all seven networks, including large networks (e.g., ResNet101), the reconstructed networks perform almost as well as their original versions. We also conduct similar experiments on ImageNet and report the results in Table S1 of Appx. B.2.

**GPU transferability.**   Our proposed attack requires the adversary to have the same brand/version of GPU as the victim, but not necessarily the same physical copy (see Fig. S3). Here, we obtain two copies of an Nvidia GTX-1080 GPU running on two different machines, using one to generate training data and another one for black-box reconstruction. We demonstrate that in this setting the models can still be well reconstructed. The experiment details and results are described in Appx. B.3.

## 5.2   Transfer Attack

To demonstrate a potential exploit of this side-channel exposure, we use reconstructed networks to launch adversarial transfer attack. Transfer attack relies on a surrogate model, one that approximates the target model, to craft adversarial examples of the target model. In a black-box setting, it is known to be hard in general case to find an effective surrogate model (Demontis et al., 2019). But under our threat model, the adversary can recover the network structure of a black-box model from the leaked magnetic signals, and use it as the surrogate model.

Here we test on six networks found in the wild, ranging from VGGs to AlexNet to ResNets (listed in Table 3). For each network (and each column in Table 3), we treat it as a black-box model and reconstruct its structure by running it on four different GPUs (listed in Table 3), obtaining four reconstructed versions. Afterwards, we train each reconstructed network and use it to craft adversarial examples for transfer attacking the original network. The transfer attack success rates are reported in the top four rows of Table 3, which are compared against several baselines shown in the bottom half of the table. Using our side-channel-based surrogate model, the transfer attack success rate is comparable to the *white-box* transfer attack baseline, that is, using the target model's network structure (but trained separately) to attack the target model itself. In other words, our side-channel-based reconstruction effectively turn a black-box attack into a white-box attack.

We also conducted additional experiments for transfer attacks on MNIST dataset. We reconstruct network models downloaded online and then launch attacks. The results are reported in Appx. B.4.

## 5.3   Discussion: Defenses Against Magnetic Side Channel Leakage

At this point, we have shown the robustness and accuracy of the magnetic side channel exploits. Traditionally countermeasures fall under the category of either prevention, detection, or jamming.

Table 3: Transfer attack results on CIFAR-10.

|  |  | Target Model | | | | | |
|---|---|---|---|---|---|---|---|
|  |  | ResNet-18 | ResNet-34 | ResNet-101 | VGG-11 | VGG-16 | AlexNet |
| Source Model | GTX-960 | 98.56 | 92.51 | 91.20 | 63.41 | 72.57 | 58.90 |
|  | GTX-1080 | 97.88 | 90.86 | 86.24 | 64.69 | 55.19 | 56.83 |
|  | TITAN-X | 98.32 | 93.45 | 84.47 | 61.89 | 77.36 | 68.41 |
|  | TITAN-V | 98.48 | 93.65 | 91.27 | 64.39 | 72.77 | 60.17 |
|  | VGG-11 | 65.86 | 57.82 | 57.52 | 60.24 | 65.50 | 39.95 |
|  | VGG-16 | 74.00 | 61.54 | 54.23 | 41.60 | 74.29 | 29.57 |
|  | ResNet-18 | 97.70 | 90.72 | 80.27 | 47.98 | 86.64 | 30.56 |
|  | ResNet-34 | 97.21 | 92.46 | 82.30 | 51.42 | 85.60 | 32.34 |
|  | ResNet-101 | 92.53 | 86.98 | 92.95 | 53.98 | 83.04 | 30.55 |
|  | AlexNet | 10.11 | 9.59 | 10.19 | 11.60 | 10.42 | 62.70 |

Since our approach is passive in that it does not alter any code or hardware operation of GPUs, detection methods which consist of somehow discovering someone is listening to the GPU are not applicable to magnetic leakage. Here we focus on *prevention* and *jamming*.

**Prevention** As shown in Figure 1, each rise and drop of the magnetic signals correspond to the boundary between GPU operations. This is only possible when the input batch is large enough to keep every GPU operation sustained and stabilized at a high-load state. To prevent this behavior, one can keep the input sufficiently small (e.g. 1 single image) such that the magnetic signals never reach any stable state and suffer from a low signal-to-noise ratio, rendering our sensing setup futile. Another way to prevent magnetic side channel leakage is to use a non-standard framework for inference which the adversary does not have any training data to start with.

**Jamming** While running on tiny input batch size might be infeasible for large input dataset, we find jamming also an effective defense mechanism. Specifically, during the inference of a large input batch, we ran a third-party CUDA GPU stress test in the background (Timonen, 2020). We found that the magnetic signals are completely distorted because of the constantly high utilization of GPU. Moreover, we observe little speed degradation for the foreground inference. The main caveat with this approach is higher power consumption and the possible effects on the lifetime of a GPU.

Another possible defense mechanism results from the fact that we are not tracking the actual dataflow in the GPU. For example, we can correctly identify two GPU operations, convolution and batch norm, in a long sequence. But there is no evidence proving the dataflow follows the same pattern—the output from convolution could be a deadend and batch norm takes input from a previous GPU operation. This mismatch between the dataflow and the underlying network model makes it hard to decipher robustly. While this defense can handle arbitrary input and networks in theory, we are unsure about the implementation hurdles of this defense and how modern deep learning libraries optimize for such unconventional graphs.

## 6 CONCLUDING REMARKS

We set out to study what can be learned from passively listening to a magnetic side channel in the proximity of a running GPU. Our prototype shows it is possible to extract both the high-level network topology and detailed hyperparameters. To better understand the robustness and accuracy, we collected a dataset of magnetic signals by inferencing through thousands of layers on four different GPUs. We also investigated how one might use this side channel information to turn a black-box attack into a white-box transfer attack.

**Limitations.** In our formulation, we assume networks progress in a linear fashion and do not handle complex graph network with intricate branching topologies. We cannot tell if a network is trained with dropout since dropout layers do not appear at inference time. Indeed, any operation that only appears during training is beyond the capability of magnetic side channel snooping.

Our reconstruction DNNs require knowledge of the victim's GPU model and version. When these are unknown, the adversary may still exhaustively pretrain multiple sets of reconstruction DNNs for all GPU models and at runtime scan through all reconstruction options. Software upgrades, which can lead to significant performance boost and therefore alter the emitted magnetic signals, may be

viewed as further increasing the set of possible GPU models. In our experiments, we keep all the software versions constant, including OS verion, CUDA version, and PyTorch/Tensorflow version.

**Ethical Considerations.**    Our main intention is to study the extent to which a magnetic side channel can leak information. We hope to help people understand that running deep learning inferences on GPUs can leak critical information, and in particular model architecture and hyperparameters, to nearby induction sensors. Sharing these findings creates the potential for malicious use. We introduced several viable defense mechanisms in 5.3.

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

# Supplementary Document
# Can one hear the shape of a neural network?:
# Snooping the GPU via Magnetic Side Channel

## A    BiLSTM Network Structure

Classifying steps in a network model requires taking in a time-series signal and converting it to labeled operations. The EM signal responds only to the GPU's instantaneous performance, but because the GPU executes a neural network sequence, there is rich context in both the window before and after any one segment of the signal. Some steps are often followed by others, such as pooling operations after a series of convolutions. We take advantage of this bidirectional context in our sequence to sequence classification problem by utilizing a BiLSTM network to classify the observed signal. To retrive a network's topology, we pass normalized EM values into a two-layer BLSTM network, with dropout of $0.2$ in between. From there we compute a categorical cross-entropy loss on a time-distributed output that we achieve by sliding an input window across our EM signal. This approach proves robust, and is the method used by all of our experiments, and on all GPU's tested.

The segmented output of our BiLSTM network on our extracted signal is for the most part unambiguous. Operations that follow one another (i.e. convolution, non-linear activation function, pooling) are distinct in their signatures and easily captured from the context enabled by the sliding window signal we use as input to the BiLSTM classifier. Issues arise for very small-sized steps, closer to our sensor's sampling limit. In such regions a non-linear activation may be over-segmented and split into two (possibly different) activation steps. To ensure consistency we postprocess the segmented results to merge identical steps that are output in sequence, cull out temporal inconsistencies such as pooling before a non-linear activation, and remove activation functions that are larger than the convolutions that precede them.

## B    Additional Experiments

### B.1    Using Levenshtein Distance to Measure Network Reconstruction Quality

To provide a sense of how the Levenshtein edit distance is related to the network's ultimate performance, we consider AlexNet (referred as model A) and its five variants (refered as model B, C, D, and E, respectively). The variants are constructed by randomly altering some of the network steps in model A. The Levenshtein distances between model A and its variants are 1, 2, 2, 5, respectively (see Fig. S1), and the normalized Levenshtein distances are shown in the brackets of Fig. S1. We then measure the performance (i.e., standard test accuracy) of these models on CIFAR-10. As the edit distance increases, the model's performance drops.

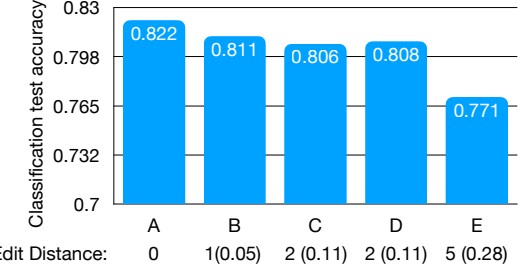

Figure S1: The model's classification accuracy drops as its Levenshtein distance from the original model (model A: AlexNet) increases.

### B.2    Reconstruction Quality on ImageNet

We treat ResNet18 and ResNet50 for ImageNet classification as our black-box models, and reconstruct them from their magnetic signals. We then train those reconstructed networks and compare their test accuracies with the original networks' performance. Both the reconstructed and original

Table S1: Model reconstruction evaluated on ImageNet classification.

| Model | ResNet18 | | ResNet50 | |
|---|---|---|---|---|
| | Original | Extracted | Original | Extracted |
| Top-1 Acc. | 64.130 | 64.608 | 62.550 | 61.842 |
| Top-5 Acc. | 86.136 | 86.195 | 85.482 | 84.738 |
| KL Div. | - | 2.39 | - | 4.85 |

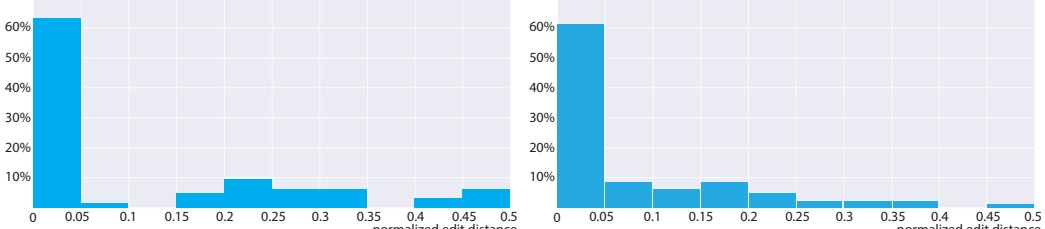

Figure S2: **Distribution of normalized Levenshtein distance.** (left) We plot the distribution of the normalized Levenshtein distances between the reconstructed and target networks. This results, corresponding to Table 1 in the main text, use signals collected on Nvidia TITAN V. (right) We also conduct similar experiments on two Nvidia GTX-1080 GPUs. One is used for collecting training signals, and the other is used for testing our side-channel-based reconstruction algorithm.

networks are trained with the same training dataset for the same number of epochs. The results are shown in Table S1, where we report both top-1 and top-5 classification accuracies. In addition, we also report a KL-divergence measuring the difference between the 1000-class image label distribution (over the entire ImageNet test dataset) predicted by the original network and that predicted by the reconstructed network. Not only are those KL-divergence values small, we also observe that for the reconstructed network that has a smaller KL-divergence from the original network (i.e., ResNet18), its performance approaches more closely to the original network.

### B.3  GPU TRANSFERABILITY

Here we verify that **(i)** the leaked magnetic signals are largely related to GPU brand/version but not the other factors such as CPUs and **(ii)** the signal characteristics from two physical copies of the same GPU type stay consistent.

We obtain two copies of an Nvidia GTX-1080 GPU running on two different machines. When we run the same network structure on both GPUs, the resulting magnetic signals are similar to each other, as shown in Fig. S3. This suggests that the GPU cards are indeed the primary sources contributing the captured magnetic signals.

Next, we use one GPU to generate training data and another one to collect signals and test our black-box reconstruction. The topology reconstruction results are shown in Table S2, arranged in the way similar to Table 1, and the distribution of normalized Levenshtein edit distance over the tested networks are shown in Fig. S2-right. These accuracies are very close to the case wherein a single GPU is used. The later part of the reconstruction pipeline (i.e., the hyperparameter recovery) directly depends on the topology reconstruction. Therefore, it is expected that the final reconstruction is also very similar to the single-GPU results.

### B.4  TRANSFER ATTACKS ON MNIST

We also conduct transfer attack experiments on MNIST dataset. We download four networks online, which are not commonly used. Two of them are convolutional networks (referred as CNN1 and CNN2), and the other two are fully connected networks (referred as DNN1 and DNN2). None of these networks appeared in the training dataset. We treat these networks as black-box models, and reconstruct a network for each of them. We then use the four reconstructed models to transfer attack the four original models, and the results are shown in Table S4. As baselines, we also use the four original models to transfer attack each other including themselves.

Table S2: Classification accuracy of network steps (GTX-1080).

|                | Prec. | Rec. | F1 | # samples |
|----------------|-------|------|------|-----------|
| LSTM | .997 | .999 | .998 | 12186 |
| Conv | .985 | .989 | .987 | 141164 |
| Fully-connected | .818 | .969 | .887 | 9301 |
| Add | .962 | .941 | .951 | 30214 |
| BatchNorm | .956 | .944 | .950 | 48433 |
| MaxPool | .809 | .701 | .751 | 1190 |
| AvgPool | .927 | .874 | .900 | 294 |
| ReLU | .868 | .859 | .863 | 11425 |
| ELU | .861 | .945 | .901 | 8311 |
| LeakyReLU | .962 | .801 | .874 | 3338 |
| Sigmoid | .462 | .801 | .585 | 5106 |
| Tanh | .928 | .384 | .543 | 8050 |
| Weigted Avg. | **.945** | **.945** | **.945** | - |

Table S3: **DNN estimation accuracies.** Using the 1804 convolutional layers in our test dataset, we measure the accuracies of our DNN models for estimating the convolutional layers' hyperparameters. Here, we break the accuracies down into the accuracies for individual hyperparameters.

|           | Kernel | Stride | Padding | Image-in | Image-out |
|-----------|--------|--------|---------|----------|-----------|
| Precision | 0.971 | 0.976 | 0.965 | 0.968 | 0.965 |
| Recall | 0.969 | 0.975 | 0.964 | 0.969 | 0.968 |
| F1 Score | 0.969 | 0.975 | 0.962 | 0.967 | 0.965 |

In Table S4, every row shows the transfer attack success rates when we use different source (surrogate) models to attack a specific original model (CNN1, CNN2, DNN1, or DNN2). Each column labeled as "extr." corresponds to the extracted (reconstructed) model whose target model is given in the previous column right before it. In addition, we also show all the models' test accuracies on MNIST in the last row of the table. The results show that all the reconstructed models approximate their targets closely, both in terms of their abilities for launching transfer attacks and their classification performance.

## C  SENSOR SETUP

The magnetic induction signal we utilize comes from digitally converting analog readings of a Texas Instruments DRV425 fluxgate sensor with Measurement Computing's USB-204 Digital Acquisition Card. The sensor samples at a frequency of 47Khz and the converter operates at 50Khz to map the originally -2mT~2mT readings across 0 to 5 Volts using a 12-bit conversion. Calibrating the sensor requires (a) that the sensor is within range of the electromagnetic signal and that (b) the sensor orientation is consistent. The magnetic induction signal falls off at a rate inversely proportional to distance squared, and so the sensor must be placed within 7mm of the GPU power cable for reliable

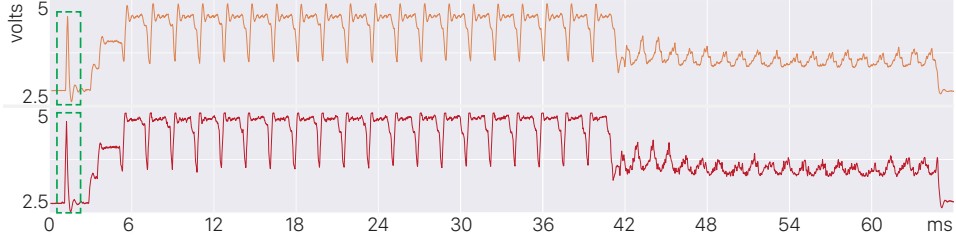

Figure S3: Here we plot the resulting signals from the same network model deployed on two different instances of a NVIDIA-GTX 1080 (running on two different computers). In the green boxes on the left are the spikes that we inject on purpose (discussed in Sec. 4.2) to synchronize the measured signal with the runtime trace of the GPU operations.

Table S4: MNIST results.

| | | Source Model | | | | | | | |
|---|---|---|---|---|---|---|---|---|---|
| | | CNN1 | extr. | CNN2 | extr. | DNN1 | extr. | DNN2 | extr. |
| Target | CNN1 | .858 | .802 | .226 | .202 | .785 | .795 | .476 | 527 |
| | CNN2 | .395 | .319 | .884 | .878 | .354 | .351 | .354 | .211 |
| | DNN1 | .768 | .812 | .239 | .223 | .999 | .999 | .803 | .885 |
| | DNN2 | .703 | .768 | .219 | .194 | .975 | .979 | .860 | .874 |
| Accuracy | | .989 | .987 | .993 | .991 | .981 | .981 | .980 | .983 |

measurement. Flipping the flat sensor over will result in a sign change of the magnetic induction signal, thus a uniform orientation should be maintained to avoid preprocessing readings across the dataset to align.

