# OpenReview forum: "Can one hear the shape of a neural network?: Snooping the GPU via Magnetic Side Channel"
_ICLR.cc/2021/Conference — Reject_

### Official Review · AnonReviewer1 · 2020-10-23
**Good paper, but may not be appropriate for ICLR**

**Rating:** 4
**Confidence:** 4

**Review:**

This paper demonstrates that magnetic side channel information from a GPU (that is processing a deep neural net) can be snooped to recover the architecture and hyperparameters of the neural network. While the concept of side channel information snooping to recover codes/software (including ML models) is widely studied, the novelty claim is that recovering detailed structures of deep models is new. The paper also demonstrates that black-box attacks mounted using a recovered model is quite powerful compared to traditional black-box attacks.

The paper is well-written and quite useful for the safety-critical applications community that use ML. However, there is no core ML contribution made in this paper. The authors use standard ML models to map the side channel signal to deep learning model architecture. The use of model architecture consistencies as constraints is clever, but nothing significant in terms of contributions to the ML community. So, in my opinion, this paper is better suited for other venues such as cyber security conferences.

---

> ### Author Response · Authors · 2020-11-15
> **AnonReviewer1 Response**
>
> We thank the reviewer for the constructive feedback. Below we provide answers to your concern.
>
> > Relevance to the ICLR community
>
> Our work follows recent published works in the machine learning community that investigate side-channel exploits used to recover network architectures in detail. Notably [1], _published in ICLR 2020_, sought to reconstruct network models from leaked side-channel information. Their approach relied entirely on non-learning computational methods to generate and prune candidate network models from their extracted trace. No novel ML method (feature representations, layer definitions, or network compositions) was claimed or introduced, rather, the novelty was the side-channel based reconstruction itself. Similarly, the reviews for [2] (submitted concurrently to ICLR 2021) _do not question the relevance of side-channel attacks to ICLR_ (here again the novelty is not in the ML method).
>
> As expressed in ICLR publications [1, 2], it is of great interest to the community to explore the hardware vulnerabilities of deployed deep learning systems. [1] argues that recovering network architectures is a good goal in its own right, and our method recovers more general and arbitrarily defined network specifications. We further progress their aims by tailoring an approach that specifically handles widely adopted GPU hardware, without installing or executing spyware; we also introduce an optimization formulation for layer parameter estimation.
>
> References:
>
> 1. Hong et al, How to 0wn the NAS in Your Spare Time, ICLR 2020
>
> 2. Private Image Reconstruction From System Side Channels Using Generative Models, ICLR 2021 submission under review

---

> > ### Author Response · Authors · 2020-11-23
> > **AnonReviewer1 Response Follow-up**
> >
> > Thank you again for your review, and please let us know if we can answer any other questions or concerns prior to the close of the discussion period.

---

### Official Review · AnonReviewer2 · 2020-10-28
**Intriguing way to reconstruct neural networks by snooping the GPU power consumption.**

**Rating:** 7
**Confidence:** 4

**Review:**

# Summary:
The paper presents a method for capturing the shape (type of layers) and their respective parameters of a neural network through the magnetic field induced as the GPU drains power. In particular, the GPU is snooped using an off-the-shelf magnetic induction sensor which is placed along the power cable of the GPU. It turns out that under some assumptions (knowledge of GPU model and deep learning framework, knowledge of input size and ability to launch query of specific batch size), based on the correlation of the power consumption pattern of the GPU with the operations being performed it is possible to recognize the type of operation being performed as well as the respective hyper-parameters with very few errors.


# Strengths:
The paper presents a very interesting and novel method for "spying" on the Neural Network being executed on a GPU by capturing the magnetic signal induced by the GPU's power consumption. A BiLSTM is employed to classify captured samplers of the magnetic signal to segments corresponding to specific steps/operations of the neural network. Then DNNs specific for each type of step are employed for estimating the hyper-parameters of each step. Consistency between the hyper-parameters is enforced by solving an integer-programming problem.

The proposed method is able to recover the entire structure of various types of neural networks (including randomly generated ones) and the parameters of each step with very few errors. The similarity of the reconstructed networks is evaluated under various metrics, including the performance of target and reconstructed networks on a classification task. An interesting application is the use of the method to build surrogate models for performing black-box adversarial attacks with very high success rates. Possible counter-measures are proposed and the limitations of the method are also discussed.

The paper is very well-written and easy to read. The evaluation is quite comprehensive showing the ability of the method to fully recover the architecture of various networks while further results and clarifications are provided in the supplemental material.


# Weaknesses and Questions:
Some interesting aspects are not completely covered. For example, although transferability to different hardware is discussed, still the sensitivity of the method with respect to some factors is not discussed. In particular, are there any calibration issues that should be taken into account? How does the method perform if a sensor of different make is used with possible different sampling rate? In what range of sampling rates/placement distances the method works? Regarding GPU transferability, are the GPU pairs of the same make, frequency settings, etc.?

Additionally, regarding the segmentation of the samples in steps, one would expect that some over-segmentation issues would be present. Are there any measures for enforcing temporal consistency and for avoiding over-segmentation, or it is not an issue because the samples are so unambiguous?

Regarding the applicability of the proposed method, in real-world conditions it may not always be possible to define the input of the batch size. On this topic, more details can be provided on the strategy to be used to find a suitable batch size for the method to work. Additionally, the paper considers only networks acting as encoders. Would the method naturally extend to decoders or are there any difficulties/ambiguities introduced? Also the paper focuses on the inference task, can the method be applied also for training tasks and with what adaptations?

Regarding the defenses, what happens if inference optimizers (e.g. Tensor RT) are used? Is the performance of the method affected? Also, does multi-GPU setups introduce interference?

## Minor comments:
* Figure 1 right: the abbreviations (BN, MP), although common, should be explained in the text/caption.
* Joint optimization paragraph: "will generally not fully"
* page 5 before Section 4: "Were it that DNN"

# Rating Justification:
Overall, I think that the paper is highly novel and introduces an interesting way to recover the architecture of a NN by using a cheap sensor. There are many questions that rise by this work mainly though due to its novel and intriguing nature. I am not 100% sure that ICLR is the most suitable venue for this work, I am not considering though this aspect in my rating.

# Rating and comments after the rebuttal
I share the concerns of fellow reviewers regarding the practicality of the assumptions needed for launching the attack presented in the paper, however, based also on the discussion with the authors, I think that the paper is interesting regardless as it can lead to better insights regarding the development of suitable defence mechanisms for securing the architecture and the information carried by a Deep Learning model.

I share also the concerns raised in the other reviews that the paper might be better appreciated by an audience focusing on cyber-security. However, I think that the subject can also be of relevance to ICLR as the paper made an effort to highlight the aspects more relevant to the ML community.

Hence, leaving the aspect of relevance to the ACs' discretion, I think that overall this is a clearly written paper based on well executed research that presents some interesting results that are potentially impactful in the aspects concerning security of systems employing Deep Learning models.

---

> ### Author Response · Authors · 2020-11-15
> **AnonReviewer2 Response**
>
> We thank the reviewer for the constructive feedback. We have updated our paper with the grammar and notation edits that were raised, and below we provide answers to your questions and concerns.
>
> > Sensitivity of Signal Extraction
>
> > > Are there any calibration issues that should be taken into account?
>
> The only sensor calibration required is to ensure (a) that the sensor is within range of the electromagnetic signal and that (b) the sensor orientation is consistent. Since the magnetic induction signal decays inversely proportional to squared distance, we must place the sensor within 7mm of the GPU power cable. Flipping the flat sensor over will result in a sign change of the magnetic induction signal, thus a uniform orientation should be maintained to avoid having to align measurements across the dataset.
> We have updated the paper to include the above clarifications in Appendix C.
>
> > > How does the method perform if a sensor of different make is used with possible different sampling rate?
>
> Previous works have attempted electromagnetic side-channel attacks with industrial probes such as the Langer RF-U 5-2 [1]. These sensors are rated to measure at higher frequencies ranging from 30MHz to 3GHz, but are far more costly (\\$1,500+ USD) and require additional expensive equipment to operate. Other power side channel exploits have explored sampling at again higher rates varying from 400KHz to 2.5GHz [2, 3]. Sampling at these frequencies allows finer capture of hardware operations that can ease network step classification, although it significantly increases the amount of feature data to be processed. By contrast, our method involves a \\$3 USD sensor sampling at 47KHz. Sampling at lower rates than 47KHz would make it difficult to adequately capture short events on the GPU such as non-linear activation functions and small matrix multiplications that may occur.
>
> > > In what range of sampling rates/placement distances does the method work?
>
> We were unable to vary the sampling rate of our sensor, and so all of our experiments use the factory 47KHz sampling frequency. Our measurements were taken at a distance of roughly 5mm from the GPUs power cable, although a reduced signal is still present up to 7-8mm away.
>
> > > Regarding GPU transferability, are the GPU pairs of the same make, frequency settings, etc.?
>
> We acquired an identical make of our NVIDIA GTX-1080 GPU and used it to extract signals with the same sampling frequency of 47KHz when verifying transferability.
>
> > Signal Segmentation
>
> The segmented output of our classification network on the extracted signal is largely unambiguous. Operations that follow one another (i.e. convolution, non-linear activation function, pooling) are distinct in their signatures and easily captured from the context learned by the BiLSTM classifier. Issues arise for very small-sized steps, closer to the sensor’s sampling limit. In undersampled regions a non-linear activation may be over-segmented and split into two (possibly different) activation steps. To ensure consistency we postprocess the segmented results to (1) merge identical steps that are output in sequence, (2) cull out pooling before a non-linear activation, and (3) remove activation functions that are larger than the convolutions that precede them. We have updated the paper to include the above clarifications in Appendix A.
>
> > Determining the Batch Size
>
> > > Regarding the applicability of the proposed method, in real-world conditions it may not always be possible to define the input of the batch size. On this topic, more details can be provided on the strategy to be used to find a suitable batch size for the method to work.
>
> We are not sure that we fully understand the question, and we would appreciate a clarification. Here we provide an answer to our best interpretation of the question.
>
> To robustly recover network topology (as opposed to hyperparameters), we do not require knowledge of batch size, only that the batch size is sufficiently large enough to provide a clear signal. While we used a \\$3 sensor with limited sampling rate and signal-to-noise ratio (SNR), a sensor with high sampling rate and SNR would correspondingly require a smaller minimum batch size.
>
> To estimate hyperparameters absent known batch size, others have considered fitting the proposed design with a known set of values using other heuristics or timing information [4, 5]; these approaches make assumptions on the family of networks within scope.
>
> On the other hand, to estimate hyperparameters for unfamiliar architectures, knowledge of input and batch size is required. We discuss two strategies here, the first of which we have tried successfully.

---

> > ### Author Response · Authors · 2020-11-15
> > **AnonReviewer2 Response (Continued)**
> >
> > A suitable batch size can be approximated if we are able to passively observe two inferences with unknown, different, but sufficiently large, batch sizes (again, “sufficiently large” depends on the quality of the sensor). The underlying principle is this: we have observed mostly linear effects of batch size on the sensor’s signal profile. Since the two inferences on the same network model have two different batch sizes, we solve for the linear rescaling (in amplitude and time) that best aligns the two observed signals. This linear rescaling acts to normalize the signal (in physics this is nondimensional analysis), and represents the observed data in a manner that has “factored out” the batch size. We found that this works in practice.
> >
> > Alternatively, a suitable batch size can be approximated if offline queries to the hidden network are permitted on similar hardware. Again, since we have observed mostly linear effects of batch size on the sensor’s signal profile, this gives us reason to believe that one could refine an approximation of the batch size from datasets generated with signals sampled coarsely at different batch sizes (e.g. 1, 32, 64, 128, 256, etc). Using these batch-labeled datasets, a DNN or KNN classifier could approximate individual parameters for each step. These estimates could then be used to run the extracted network topology at the candidate batch sizes to see which profile most closely aligns with the original signal. If necessary, the batch size estimate could be further refined by either repeating the procedure with a narrower search for batch candidates or attempting to interpolate ([6]) between two potential batch sizes.
> >
> > > Method Variations and Extensions
> >
> > > > Would the method naturally extend to decoders?
> >
> > The handling of decoders differs in the constraints used for the integer programming problem to ensure parameter consistency. Decoders are composed of the same functional steps as encoders, but potentially grow in size across steps. Treating decoders requires reassessing the optimization constraints to enforce that layers remain the same size or grow. Recovering encoder-decoder networks introduces the additional challenge of pinpointing the switch from encoder to decoder. However, if this transition can be detected or approximated, an additional boundary constraint could be introduced to the integer programming optimization that would allow the network to be split into two optimization problems constrained to align at the transition. This formulation would provide a natural extension of our method to handle encoder-decoder networks though any such attempts remains future work.
> >
> > > > Can the method be applied for training tasks?
> >
> > The method may naturally extend to training tasks by ignoring signals related to back-propagation and focusing only on the forward steps. Access to training would provide numerous examples of the network to work with, rather than the single extracted source acquired during inference, and would likely increase the recovery accuracy of our method. One could straightforwardly apply our method to every training epoch and consolidate the proposed networks using statistics and heuristics on the candidate models.
> >
> > > > What happens if inference optimizers are used?
> >
> > Inference optimizers have the effect of introducing new categories to the supervised BiLSTM classification task, either by merging commonly paired operations (i.e. fully connected step and batch norm) or introducing new operations altogether (i.e optimized mean computation for average pooling). Changing the classification categories will therefore provide a defense against a BiLSTM classifier trained without optimizers in its dataset. However, it is possible to build up a dataset consisting of network inferences where optimizers are both turned on and off, generating an encompassing BiLSTM classifier that would be robust to such defenses.
> >
> > > > Do multi-GPU setups introduce interference?
> >
> > Multi-GPU setups do not introduce interference so long as their power cables are isolated and sufficiently far apart (>7mm). Recording the signal on the GPU itself or near the power cable plugs into the GPU is an easy way to avoid interference from multi-GPU setups. We have experimented with recording multi-GPU setups and found no issues isolating the side-channel information of the victim GPU.
> >
> > References:
> > 1. Batina et al, CSI NN: Reverse engineering of neural network architectures through electromagnetic side channel, USENIX Security 2019
> > 2. Wei et al, I know What You See: Power Side-Channel Attack on Convolutional Neural Network Accelerators, ACSAC 2018
> > 3. Xiang et al, Open DNN Box by Power Side-Channel Attack, IEEE ToCaSII 2020
> > 4. Hong et al, How to 0wn the NAS in Your Spare Time, ICLR 2020
> > 5. Hu et al, Deepsniffer: A dnn model extraction framework based on learning architectural hints, ASPLOS 2020
> > 6. Jeong et al, Weighted Dynamic Time Warping for Time Series Classification, CAIP 2011

---

> > > ### Comment · AnonReviewer2 · 2020-11-23
> > > **Clarification on Batch Size**
> > >
> > > Dear authors,
> > >
> > > first of all, thank you for the clarifications provided in your response to this as well as the other reviews. Regarding my comments on how to determine a suitable Batch Size I had not fully understood the reason that a sufficiently large batch size is required. In fact, my understanding was that it is required that the GPU saturates, however, from your feedback it becomes clear that it is only a matter of achieving a sufficiently high SNR to allow for reliable recognition which is also related to the specifications of the sensor. Although in retrospect it seems clear to me, it might still be useful to further stress in the manuscript why a sufficiently large batch size is required.
> > > My comment/concern was that in some cases, based on the way the model is served, one might be "forced" to use a specific batch size e.g. one sample. Your answers though sufficiently addressed this comment.

---

> > > > ### Author Response · Authors · 2020-11-23
> > > > **Respone to Clarification**
> > > >
> > > > We greatly appreciate your response and have updated the paper to stress the relationship between the batch size and signal to noise ratio resulting from the choice of sensor.

---

### Official Review · AnonReviewer3 · 2020-10-29
**Interesting but assumptions are not practical**

**Rating:** 5
**Confidence:** 4

**Review:**

Summary:
- This paper studies the effectiveness of inferring a neural network’s layers and hyperparameters using the magnetic fields emitted from a GPU’s power cable. The results show that (under certain assumptions) one can reconstruct a neural network’s layers and hyperparameters accurately, and use the inferred model to launch adversarial transfer attacks.

Strong points:
- The idea of using magnetic side channels to infer network structure is interesting.
- The paper is well-written with ideas and limitations explained clearly.
- The experiment results are thorough and explained clearly

Weak points:
- The threat model seems impractical. Attacker assumptions include:
  - have physical access to the GPU
  - know the exact input feature dimensions and batch size.
  - know the deep learning framework, GPU brand and hardware/software versions.
- The main innovation is demonstrating magnetic side channels from GPU cables reveal information about network structures. However, I’m not sure if ICLR is the best venue for this type of contribution. This paper could be a much stronger submission to other security and system conferences.

Recommendation:
- I’m inclined to recommend a reject. The main reason is that the results are based on multiple impractical assumptions, limiting the impact of this paper in reality.

Comments & questions:
- How do the authors imagine launching this attack in reality? Specifically, how would one know the input dimensions and batch size of a black-box model? A clear explanation of this will help readers understand the value of this work.
- Using consistency constraints to optimize for hyperparameter estimation is interesting. How effective is this additional optimization compared with only using the initial estimation?

Minor comments
- “But there is ‘not’ evidence proving” -> ‘no’

==== Updates after the response ====

I thank the authors for answering the questions in detail. Providing an example application does help readers understand scenarios where the threat model could apply. However, I still think such scenarios are not common but agree that the findings in this paper could be helpful for future security research. I adjusted my rating based on this better understanding.

---

> ### Author Response · Authors · 2020-11-15
> **AnonReviewer3 Response (Part1)**
>
> We thank the reviewer for the constructive feedback. We have updated our paper to remedy the grammar and clarification edits that were raised, and below we provide answers to your questions and concerns.
>
> > Clarification of Example Attack and Motivation
>
> > > Q: How do the authors imagine launching this attack in reality? Specifically, how would one know the input dimensions and batch size of a black-box model? A clear explanation of this will help readers understand the value of this work.
>
> For the specific question: Many side-channel attacks [1,2,3,4,5] involve running queries. If one can run a query, it means they can control the batch size; it also means they know the input dimensions. Since running queries is not unusual for side-channel attacks, it follows that knowledge of input dimensions and batch size are not unusual either.
>
> Stepping back to the broader question: to set the stage, let us paint one concrete application. Imagine renting a vertically integrated internal cloud platform for ML (e.g., IBM Cloud Pak for Data with Watson Assistant deployed on local hardware [6]). Such platforms are designed to be tamper resistant: users should be able to access services via an application interface (API), but not have access to easily reverse engineer details of the vendor’s proprietary design/architectures. While a user may be motivated to steal and replicate the design, their access is limited to the API and physical access, as opposed to “virtual” (logic/software/programming) access.
>
> Most broadly, the threat model presented in the paper (page 1) offers interesting strengths and weaknesses compared to existing models. Because GPUs are now widespread in deep learning, having a variety of non-dominating threat models is useful. Compared to other attacking methods [7, 8], magnetic induction side-channel attacks do not require any code to run on the host machine in order to alter or leak logic. This makes electromagnetic side-channel attacks particularly attractive in cases where the attacker has no permission or opportunity to access software. These scenarios may lead to a false sense of security, since magnetic side channel exploits are pervasive and difficult to both detect and defeat [3]. Merely drawing power to perform operations can lead to changes in the nearby magnetic field for electricity driven hardware, making magnetic side-channel access to GPUs intrinsic. Our work explores the extent to which electromagnetic side-channels can leak network information in these settings.
>
> > Optimization vs Initial Estimate
>
> In Table S3 of the supplemental document, we show that initial hyperparameter estimates are quite accurate, when considering each in isolation. However, our goal is to recover a _valid_ architecture: layers/steps must have compatible dimensions and hyperparameters. Without optimization, we found that for all but the shallowest networks, one or more inconsistencies in the estimated hyperparameters invalidated the architecture. Therefore, the optimization is not only highly effective, but is indeed a strictly necessary step to ensure recovery of a fully valid architecture.
>
> References:
>
> 1. Xiang et al, Open DNN Box by Power Side-Channel Attack, IEEE ToCaSII 2020
>
> 2. Wei et al, I know What You See: Power Side-Channel Attack on Convolutional Neural Network Accelerators, ACSAC 2018
>
> 3. Han et al, Watch Me, but Don’t Touch Me! Contactless Control Flow Monitoring via Electromagnetic Emanations, ACM SIGSAC 2017
>
> 4. Batina et al, CSI NN: Reverse engineering of neural network architectures through electromagnetic side channel, USENIX Security 2019
>
> 5. Hua et al, Reverse Engineering Convolutional Neural Netowrks Through Side-channel Information Leaks, ACM DAC 2018
>
> 6. IBM Cloud Pak for Data, https://newsroom.ibm.com/2019-10-21-IBM-Advances-Watson-Anywhere-with-New-Clients-and-Innovations-Designed-to-Make-it-Even-Easier-to-Scale-AI-Across-Any-Cloud
>
> 7. Hong et al, How to 0wn the NAS in Your Spare Time, ICLR 2020
>
> 8. Hu et al, Deepsniffer: A dnn model extraction framework based on learning architectural hints, ASPLOS 2020

---

> > ### Author Response · Authors · 2020-11-15
> > **AnonReviewer3 Response (Part2)**
> >
> > > Practicality of Threat Model
> > > > Physical access to the GPU
> >
> > Our approach trades virtual (in-execution) access for physical access. Some threat models allow for significant virtual invasiveness (installing spyware [7, 8]) but no physical access, whereas other methods require physical access to measure physical variations near microcontrollers [3, 4]. Each methodology offers its own advantages, and either can be seen as "more practical" depending on the situation.
> >
> > Physical side-channels can be accessed regardless of encryption or user-permissions, but the leaked information must be observed locally. Attacks based on physical proximity are practical and prejudicial, leading to insights that may be garnered on one more easily-accessed device and transferred to other physically-secured devices that deploy the same models or network architectures [8]. Software based side-channels trade off physical access for the ability to install and run processes on the host machine, at times even going as far as requiring that victim programs manipulate the same level cache as added spyware [7]. This trade-off between virtual and physical access is interesting to explore and expands the overall exposure to vulnerabilities.
> >
> > > > Knowledge of exact input feature dimensions and batch size
> >
> > As mentioned previously, many side-channel attacks [1,2,3,4,5] involve running queries, and almost all black-box attacks involve running queries. If I can run a query, it means I can control the batch size; it also means I know the input dimensions. Since running queries is not unusual for side-channel attacks, it follows that knowledge of input dimensions and batch size are not unusual either.
> >
> > > > Knowledge of the deep learning framework, GPU brand and hardware/software versions
> >
> > From a practical standpoint, GPU hardware information is already known by virtue of the physical side channel attack requiring local access to the computer. However, knowledge of the underlying hardware is not an assumption of physical side-channel attacks alone. At times even _software access_ side-channel attacks (which for example depend on memory cache) must ensure their code is operating on the same physical hardware as the host. In fact, even for software access side-channel attacks, to validate the recovered side-channel information, it is often necessary to have local access to similar hardware used to profile and compare results [7]. As shown in our GPU transferability experiments, it is _not_ necessary for our method to perform training on the same victim hardware. Datasets can be collected offline on similar hardware, as suggested in [7], to collect signal proxies of how steps will appear in practice.
> >
> > > > Software framework
> >
> > As for the software framework, the number of choices are limited. In the same manner as [7], there is no limitation that restricts which software may be used, only that the framework is publicly available so that offline training and analysis of the side-channel is possible. Therefore it is feasible and tractable to iterate through popular software candidates [7], such as PyTorch and Tensorflow, to collect samples of all the candidate signals when composing the training dataset.
> >
> > References:
> >
> > 1. Xiang et al, Open DNN Box by Power Side-Channel Attack, IEEE ToCaSII 2020
> >
> > 2. Wei et al, I know What You See: Power Side-Channel Attack on Convolutional Neural Network Accelerators, ACSAC 2018
> >
> > 3. Han et al, Watch Me, but Don’t Touch Me! Contactless Control Flow Monitoring via Electromagnetic Emanations, ACM SIGSAC 2017
> >
> > 4. Batina et al, CSI NN: Reverse engineering of neural network architectures through electromagnetic side channel, USENIX Security 2019
> >
> > 5. Hua et al, Reverse Engineering Convolutional Neural Netowrks Through Side-channel Information Leaks, ACM DAC 2018
> >
> > 6. IBM Cloud Pak for Data, https://newsroom.ibm.com/2019-10-21-IBM-Advances-Watson-Anywhere-with-New-Clients-and-Innovations-Designed-to-Make-it-Even-Easier-to-Scale-AI-Across-Any-Cloud
> >
> > 7. Hong et al, How to 0wn the NAS in Your Spare Time, ICLR 2020
> >
> > 8. Hu et al, Deepsniffer: A dnn model extraction framework based on learning architectural hints, ASPLOS 2020

---

> > > ### Author Response · Authors · 2020-11-23
> > > **AnonReviewer3 Response Follow-up**
> > >
> > > Thank you again for your review, and please let us know if we can answer any other questions or concerns prior to the close of the discussion period.

---

### Author Response · Authors · 2020-11-23
**Summary of Our Responses**

We thank our reviewers again for taking the time to read, evaluate our work, and provide constructive feedback.

We have uploaded a revised version of our paper, with edits to address the concerns raised.
Here, we summarize our responses and updates below:

> Reviewer 1

We provide our answer to the concerns regarding relevance to the community in light of recent works submitted and accepted to ICLR (in our comments).

> Reviewer 2

We discuss details related to our choice of sensor and the sensitivity of our extracted signal, adding these clarifications to the Appendix of our revised paper.

Our comments further suggest alternative step-parameter approximation schemes if other assumptions may be made, and explore the extensions and variations to our current approach proposed by the reviewer.

We have updated the paper to stress the relationship between the batch size and signal to noise ratio resulting from the choice of sensor.

>Reviewer 3

We clarify the attack motivation in the context of both recent works and real world use cases.

We address the importance of using our optimization scheme to ensure valid architectures without having to restrict step specifications to predefined parameter sets.

Lastly we review the practicality of our attack assumptions in light of recently published works on side-channel analysis, including tradeoffs between physical and virtual access to a target computer.

&nbsp;

Please see our replies to each reviewer for our detailed responses to individual points.

---

### Decision · Program_Chairs · 2021-01-07
**Final Decision**

**Decision:**

Reject

**Comment:**


The paper presents a side-channel attack in a scenario where the attacker is able to place a induction sensor near the power cable of the victim's GPU. The authors train a neural network to analyse the magnetic flux measured by the sensor to recover the structure (layer type and layer parameters) of the target neural network. The authors also show that for a wide range of target network structure, by training a network with the inferred structure, they produce adversarial examples as effective as a white box attack.

The points raised by the reviewers were the following: 1) the result that this type of side-channel attack works is interesting, 2) the practicality of the attack is unclear because the attacker needs hardware access to the victim's GPU, 3) the ML contribution is not really clear and a venue on cyber-security might be more appropriate.

Side-channel attacks on deep neural networks can be of relevance to ICLR (as pointed to by the authors by the ICLR papers/submissions on system side-channel attacks). Nonetheless, I tend to agree with R1 and R2 that the ML contribution is limited (either in terms of application of ML or methodology), and the concerns of practicality of the approach make me lean towards rejection.